# Innovative Biotherapies and Nanotechnology in Osteoarthritis: Advancements in Inflammation Control and Cartilage Regeneration

**DOI:** 10.3390/ijms252413384

**Published:** 2024-12-13

**Authors:** Binhan Liu, Tao Liu, Yanhong Li, Chunyu Tan

**Affiliations:** Department of Rheumatology and Immunology, West China Hospital of Sichuan University, Chengdu 610041, China; liubinhan666@163.com (B.L.); 15732193090@163.com (T.L.)

**Keywords:** osteoarthritis, biotherapies, regenerative medicine, inflammation, nanotechnology, cartilage repair

## Abstract

Osteoarthritis (OA) is among the most prevalent degenerative joint disorders worldwide, particularly affecting the aging population and imposing significant disability and economic burdens. The disease is characterized by progressive degradation of articular cartilage and chronic inflammation, with no effective long-term treatments currently available to address the underlying causes of its progression. Conventional therapies primarily manage symptoms such as pain and inflammation but fail to repair damaged tissues. Emerging biotherapies and regenerative medicine approaches offer promising alternatives by addressing cartilage repair and inflammation control at the molecular level. This review explores the recent advancements in biotherapeutic strategies, including mesenchymal stem cell (MSC) therapy, growth factors, and tissue engineering, which hold the potential for promoting cartilage regeneration and modulating the inflammatory microenvironment. Additionally, the integration of nanotechnology has opened new avenues for targeted drug delivery systems and the development of innovative nanomaterials that can further enhance the efficacy of biotherapies by precisely targeting inflammation and cartilage damage. This article concludes by discussing the current clinical applications, the ongoing clinical trials, and the future research directions necessary to confirm the safety and efficacy of these advanced therapies for OA management. With these advancements, biotherapies combined with nanotechnology may revolutionize the future of OA treatment by offering precise and effective solutions for long-term disease management and improved patient outcomes.

## 1. Introduction

Osteoarthritis (OA) is a widespread degenerative joint disorder characterized by the progressive deterioration of articular cartilage, accompanied by changes in subchondral bone and inflammation of the synovium [1]. It is a leading cause of disability in older adults, affecting approximately 10% of men and 18% of women over 60 years of age worldwide [2]. The growing global population of elderly individuals, rising obesity rates, and increased incidence of joint injuries have amplified the public health impact of OA, which affects over 250 million people globally [3,4].

Existing treatments, such as physical therapy, anti-inflammatory drugs, and intra-articular injections, focus on alleviating symptoms but fail to address the underlying disease mechanisms [5]. Consequently, developing therapeutic strategies that target long-term management and functional recovery remains critical. OA’s key pathogenic processes, chronic inflammation, and cartilage destruction, are central to new therapeutic interventions. Recent advances in biological therapies have shown potential in modulating inflammation and promoting cartilage repair.

Biological treatments, including cell-based therapies and the use of growth factors, aim to mitigate inflammation and stimulate tissue repair. Meanwhile, regenerative medicine focuses on restoring damaged tissues’ function through tissue repair or regeneration [6,7,8,9]. Nanotechnology has also emerged as a promising tool in regenerative medicine, offering potential benefits in anti-inflammatory applications, cartilage repair, and early OA diagnosis, thus providing a new dimension in effective disease management [10,11,12].

This review will explore the recent progress in biotherapies and regenerative medicine for OA, particularly emphasizing inflammation control and nanotechnology. This review will examine the pathophysiology of OA, current treatment challenges, and the latest therapeutic innovations, including cell therapies, growth factors, tissue engineering [13], and nanotechnology. It concludes by discussing clinical prospects and future research directions, emphasizing further studies to confirm safety, efficacy, and precise long-term management strategies [14].

## 2. Pathological Mechanisms and Emerging Treatments for OA

### 2.1. Pathogenesis and Inflammatory Mechanisms in OA

OA is a degenerative joint disease characterized by the gradual breakdown of articular cartilage, inflammation of the synovium, and remodeling of the subchondral bone. Clinically, OA manifests as pain, swelling, stiffness, and joint instability, often accompanied by joint space narrowing visible on radiographic imaging (Figure 1). While traditionally considered a result of mechanical wear and tear, inflammation plays a central role in OA progression [15,16].

In OA, synovial tissue is infiltrated by inflammatory cells, including macrophages and T and B lymphocytes. These immune cells secrete pro-inflammatory cytokines, such as tumor necrosis factor alpha (TNF-α), interleukin-1 beta (IL-1β), and interleukin-6 (IL-6) [17]. Exposure to these cytokines induces synoviocytes and chondrocytes to produce mediators like prostaglandin E2 (PGE2), nitric oxide (NO), and matrix metalloproteinases (MMPs), which contribute to cartilage degradation [18].

IL-1β is a pivotal mediator in OA pathogenesis, activating intracellular signaling pathways, such as inducible nitric oxide synthase (iNOS), cyclooxygenase-2 (COX-2), and the nuclear factor kappa light chain-enhancer of activated B cells (NF-κB). These pathways drive chondrocyte apoptosis and upregulate MMPs, especially MMP-13, a key enzyme responsible for degrading type II collagen in cartilage [19,20,21]. These processes accelerate cartilage degradation and exacerbate OA progression.

When activated, macrophages contribute significantly to OA pathogenesis by differentiating into osteoclasts that degrade subchondral bone. Studies indicate that macrophage accumulation in the knee joints of OA patients correlates with disease severity, including pain and reduced joint space [22,23]. These macrophages release B-cell activating factor (BAFF), promoting pro-inflammatory Th1 and Th17 responses while suppressing anti-inflammatory Th2 responses, further driving inflammation and tissue destruction in advanced OA [24,25].

Oxidative stress also plays a pivotal role in OA by generating reactive oxygen species (ROS) that damage chondrocytes and the cartilage matrix. This oxidative damage accelerates cartilage degradation and perpetuates inflammation, further linking ROS with key signaling pathways in OA pathology [26,27]. Antioxidant defense mechanisms are thus essential in mitigating this oxidative stress.

Recent studies have highlighted the contribution of extracellular vesicles (EVs) released by synovial cells in OA pathogenesis. These EVs transport bioactive molecules, including non-coding RNAs like miR-126-5p, which promote chondrocyte apoptosis and cartilage degradation via pathways such as PGC1α-BNIP3 mitochondrial autophagy [28,29]. Additionally, systemic low-grade inflammation, which is often associated with aging, obesity, and metabolic syndrome, exacerbates OA progression [30].

### 2.2. Biotherapy in OA: Cellular and Molecular Approaches

Biological therapies for OA encompass stem cell therapy, gene therapy, biological agents, growth factor therapy, platelet-rich plasma (PRP) therapy, and cytokine modulation. These therapies aim to reduce inflammation and facilitate cartilage repair by targeting disease mechanisms that slow progression, alleviate symptoms, and restore joint function (Figure 2). Key components include biologic agents, such as monoclonal antibodies, that inhibit pro-inflammatory cytokines like TNF-α, IL-1, and IL-6, and cytokine inhibitors, such as IL-1 receptor antagonists and TNF inhibitors, which modulate inflammation [31,32,33].

Stem cell therapies, particularly those involving mesenchymal stem cells (MSCs) and induced pluripotent stem cells (iPSCs), hold significant promise due to their regenerative and anti-inflammatory properties. MSCs promote cartilage repair and modulate immune responses through their differentiation potential and the secretion of bioactive molecules [34,35,36]. Gene therapy techniques, including CRISPR-Cas9, enable precise genetic modifications to enhance anti-inflammatory activity and tissue regeneration [37]. Additionally, exosome therapy, which utilizes stem cell-derived exosomes, offers a novel approach to modulating inflammation and promoting chondrocyte regeneration [28]. Combining these therapies with three-dimensional scaffolds in tissue engineering opens possibilities for cartilage regeneration, employing materials such as hyaluronic acid and collagen to protect and repair joints [6,38].

#### 2.2.1. Recent Breakthroughs in MSC Therapy

Recent advancements in MSC therapy have demonstrated their potential in treating OA. Sources of MSCs include bone marrow, adipose tissue, and synovial tissue. MSCs can migrate to injury sites, suppress inflammation, and enhance tissue regeneration via paracrine signaling. Clinical trials have shown MSCs’ effectiveness in alleviating pain, improving joint function, and promoting cartilage regeneration [1]. MSCs derived from different tissues show varied efficacy in treating OA. For example, bone marrow-derived MSCs (BM-MSCs) are favored for their differentiation potential, while adipose-derived MSCs (AD-MSCs) are recognized for their anti-inflammatory properties [15,23]. Moreover, combining MSCs with PRP or EVs enhances therapeutic outcomes, leveraging the regenerative properties of MSCs and the signaling capabilities of EVs [7,24,39].

However, challenges remain, such as low cell survival rates and limited bioactivity in diseased tissues. Researchers are developing advanced hydrogel systems that improve MSC viability and therapeutic efficacy [40]. Gene editing technologies like CRISPR/Cas9 also present opportunities to enhance MSC functionality and stability for better therapeutic outcomes [37].

MSCs have garnered significant attention for their potential in treating OA. MSCs exhibit their therapeutic effects through both direct differentiation into chondrocytes and robust paracrine signaling, releasing bioactive molecules like transforming growth factor beta (TGF-β), vascular endothelial growth factor (VEGF), and interleukin-10 (IL-10) [41]. These factors modulate inflammation, inhibit cartilage degradation, and promote extracellular matrix (ECM) synthesis [42].

Studies have shown that MSCs can regulate immune responses by polarizing macrophages from pro-inflammatory M1 to anti-inflammatory M2 phenotypes, thus reducing synovial inflammation and protecting cartilage tissue [43]. Furthermore, MSC-derived exosomes carry microRNAs (e.g., miR-140-3p and miR-21) that directly target apoptotic and inflammatory pathways, enhancing cartilage repair.

For instance, clinical trials have demonstrated that the intra-articular injection of MSCs resulted in significant pain relief and functional improvement in patients with knee OA, particularly when combined with PRP or EVs [44]. Despite these advancements, challenges such as low cell retention rates and limited engraftment remain critical barriers to clinical efficacy [45].

Preclinical studies have also indicated that combining MSCs with nanotechnology-based carriers improves therapeutic outcomes. Nanoparticles can shield MSCs from the harsh joint microenvironment, enhance their retention in the cartilage, and enable the sustained release of bioactive molecules [46]. These advances highlight the potential of MSCs as a cornerstone in OA biotherapy, but further clinical studies are needed to establish their long-term safety and efficacy [47].

#### 2.2.2. Role of Growth Factors and Cytokines in Cartilage Repair and Regeneration

Cartilage repair is a complex process requiring the precise coordination of various cellular and molecular mechanisms, with growth factors and cytokines playing pivotal roles in maintaining the delicate balance necessary for effective repair. Growth factors such as transforming growth factor beta (TGF-β), bone morphogenetic proteins (BMPs), and fibroblast growth factor (FGF) are known to significantly promote the proliferation and differentiation of chondrocytes, the specialized cells responsible for cartilage formation. On the other hand, cytokines like interleukin-1 (IL-1) and tumor necrosis factor alpha (TNF-α) play a regulatory role, influencing both anabolic and catabolic processes that are vital for cartilage maintenance and repair [48]. The delicate balance between these growth factors and cytokines is essential for maintaining cartilage homeostasis, ensuring that the cartilage remains healthy and functional. Recent advances in understanding the molecular pathways influenced by these critical molecules have paved the way for innovative strategies to enhance cartilage regeneration. These developments offer promising prospects for treating degenerative joint diseases, providing hope for improved outcomes in patients suffering from such conditions.

### 2.3. Regenerative Medicine and Tissue Engineering Strategies for Cartilage Repair

Regenerative medicine focuses on repairing or regenerating damaged tissues by leveraging advances in stem cell research, tissue engineering, gene therapy, and biomaterials. In the context of OA, regenerative strategies emphasize cartilage repair, inflammation control, and biomaterial applications.

#### 2.3.1. Advances in Tissue Engineering for Cartilage Regeneration

Tissue engineering combines cells, supporting structures (degradable matrices or scaffolds), and biomolecules, such as growth factors, to develop constructs that mimic natural cartilage, facilitating repair. Gene editing technologies like CRISPR-Cas9 have enhanced chondrocyte properties by reducing inflammatory signals. For instance, knocking out the TGF-β-activated kinase 1 (TAK1) gene in chondrocytes has been shown to confer anti-inflammatory effects and promote cartilage regeneration [5]. Autologous matrix-induced chondrogenesis (AMIC), a cell-free technique, also supports cartilage regeneration by releasing MSCs from bone marrow through microfracture. Studies have demonstrated AMIC’s safety and effectiveness in treating full-thickness cartilage defects [49,50,51].

#### 2.3.2. Role of Biomaterials and Scaffolds in Cartilage Repair

Biomaterials and scaffolds are crucial in cartilage restoration, providing a three-dimensional framework that facilitates cell adhesion, proliferation, and differentiation. Research by Velasco-Salgado et al. suggests that polymeric biomaterials can stimulate local cells to repair damaged cartilage and restore its viscoelastic properties [6]. Moreover, injectable hydrogels have garnered significant attention as a novel therapeutic approach due to their ability to emulate the properties of natural cartilage and be directly administered to the injury site minimally invasively.

The investigation by Atwal et al. demonstrated that these hydrogels can serve as effective carriers for bioactive molecules, promoting cartilage repair and regeneration [52]. Additionally, Kayakabe et al. utilized hyaluronic acid gel sponges as carriers to successfully transplant bone marrow-derived MSCs into rabbit joints, effectively restoring damaged articular cartilage [53]. Haleem et al. suggested that the intra-articular implantation of fibrin gel cell scaffolds containing autologous BM-MSCs and PRP may offer an effective treatment strategy for repairing articular cartilage damage [54].

Low-intensity pulsed ultrasound (LIPUS) combined with MSC transplantation has also shown potential in enhancing cartilage regeneration. Research by Chen et al. indicated that LIPUS significantly promotes the chondrogenic differentiation of MSCs by inhibiting the tumor necrosis factor (TNF) signaling pathway, thereby facilitating cartilage regeneration [55].

Incorporating nanotechnology into scaffold materials has greatly improved the delivery of growth factors and anti-inflammatory agents, accelerating cartilage regeneration. For instance, researchers have employed cationic nanocarriers to effectively transport rhein (RH) into the cartilage matrix, establishing a drug reservoir that significantly diminishes inflammatory responses and oxidative stress, thus encouraging cartilage regeneration [56]. Furthermore, collagen and xanthan gum have been used as bio-ink to fabricate tissue structures with vascular networks using 3D bioprinting technology. These structures exhibited excellent stability and biocompatibility in in vitro cultures [57].

In addition, ECM scaffold materials modified with antimicrobial peptides have demonstrated exceptional antibacterial properties and tissue repair capabilities, effectively preventing infections and promoting wound healing [58]. These studies underscore that the innovative application of novel biomaterials and scaffolds offers new opportunities for cartilage repair, highlighting their immense potential in regenerative medicine. This progress broadens the options for tissue engineering treatments of OA and brings renewed hope for improved patient outcomes.

### 2.4. Nanotechnology Innovations in OA Treatment

Nanotechnology has gained increasing attention in OA treatment, particularly in controlling inflammation and promoting cartilage regeneration. Since inflammatory responses and cartilage degradation characterize OA, nanotechnology offers novel therapeutic approaches targeting these key pathological mechanisms (Figure 3).

#### 2.4.1. Controlling Inflammatory Pathways with Nanotechnology

Nanoparticles can effectively modulate inflammation by inhibiting inflammatory mediators, modulating immune cell functions, and enhancing tissue repair. By targeting pathways such as NF-κB, nanoparticles help reduce arthritis symptoms and prevent cartilage degradation [59]. Nanoparticles have also shown the ability to shift macrophage polarization from pro-inflammatory M1 to anti-inflammatory M2 phenotypes, thereby reducing synovial inflammation and joint damage [8,60]. Additionally, nanoparticles loaded with anti-inflammatory drugs have decreased inflammatory cell infiltration and cytokine secretion, thereby protecting cartilage and slowing disease progression [61]. This approach has proven beneficial in preclinical studies and early-stage clinical research, paving the way for future therapeutic applications.

Nanotechnology provides a versatile platform for drug delivery in OA therapy by enabling the targeted modulation of inflammatory pathways. Nanoparticles play a critical role in regulating key inflammatory pathways, including NF-κB and MAPK, which are central to OA progression. Gold nanoparticles (AuNPs) have been shown to inhibit NF-κB activation, reducing the production of pro-inflammatory cytokines such as TNF-α and IL-1β. This suppression prevents cartilage degradation and chondrocyte apoptosis, significantly improving joint function in preclinical models [46]. Similarly, polymeric nanoparticles loaded with dexamethasone effectively target the MAPK signaling pathway, decreasing the expression of matrix metalloproteinases (MMPs). By mitigating cartilage matrix breakdown, these nanoparticles promote tissue repair and provide a promising therapeutic strategy for OA [62].

Nanoparticles (NPs) offer a targeted approach to modulating inflammatory pathways in OA. For example, NPs loaded with interleukin-1 receptor antagonist (IL-1Ra) have been shown to suppress synovial inflammation by inhibiting the IL-1β signaling cascade [63]. Similarly, gold nanoparticles (AuNPs) modulate the NF-κB signaling pathway, preventing chondrocyte apoptosis and reducing cartilage degradation [64].

Additionally, nanoparticles functionalized with cell-penetrating peptides, such as RGD peptides, enhance binding to integrin receptors on chondrocytes, enabling the precise delivery of anti-inflammatory agents [65]. Studies have also demonstrated that polyethylene glycol (PEG)-coated nanoparticles increase circulation time and reduce immune clearance, improving bioavailability and therapeutic outcomes [66,67]. PEG-coated nanoparticles also play a significant role in immune modulation by shifting macrophages from the M1 phenotype to the anti-inflammatory M2 phenotype. This polarization reduces synovial inflammation and decreases the secretion of pro-inflammatory cytokines like TNF-α and IL-6, thereby promoting cartilage repair and joint health [68,69].

The synergistic use of MSC-derived exosomes and nanoparticles has also shown promise in preclinical studies. Combined delivery systems enhanced anti-inflammatory effects and cartilage repair compared to single therapies [70]. For instance, combining MSC-derived exosomes with nanoparticles carrying TGF-β demonstrated superior cartilage regeneration in animal models [71]. Moreover, nanoparticles loaded with IL-1Ra or functionalized with RGD peptides have demonstrated superior binding to inflamed cartilage and prolonged retention in joint tissues. These systems have significantly reduced inflammatory cytokines and improved joint function in animal models, paving the way for clinical translation [42,72].

Despite these advancements, challenges such as ensuring nanoparticle biocompatibility and minimizing off-target effects remain critical. Further research is needed to optimize nanoparticle formulations to improve their specificity, therapeutic efficacy, and safety profiles. Innovations in nanoparticle surface modifications, such as dual-functionalized systems, hold promise for enhancing tissue targeting and achieving better outcomes in OA therapy. Future research should focus on optimizing nanoparticle formulations to improve their specificity and therapeutic efficacy in OA.

#### 2.4.2. Nanomaterials for Targeted Drug Delivery and Cartilage Regeneration

Nanotechnology enables the precise control and manipulation of materials at the nanoscale, providing unique physical, chemical, and biological properties that enhance their use in drug delivery and tissue regeneration [57,58]. Nanomaterials like gold nanoparticles (AuNPs), liposomes, and polymeric nanoparticles deliver anti-inflammatory agents and growth factors directly to target sites, thereby enhancing their bioavailability and efficacy [62,63]. Functionalizing nanomaterials with ligands or peptides improves their specificity, enabling targeted delivery to articular cartilage cells or inflammatory cells, thus increasing therapeutic precision [64,65]. Surface modifications of nanoparticles significantly influence their therapeutic efficacy and bioavailability in OA treatment. Among these modifications, polyethylene glycol (PEG) coatings can extend circulation time by reducing opsonization and immune clearance, thereby improving the bioavailability of anti-inflammatory drugs [73]. Peptide functionalization, such as with RGD (arginine-glycine-aspartate), enhances binding specificity to integrin receptors on chondrocytes, thereby facilitating targeted delivery and promoting cartilage repair [74].

Liposomal encapsulation provides a protective mechanism for therapeutic agents, preventing premature degradation and enabling localized delivery while minimizing systemic toxicity [75]. These advanced surface modifications enhance targeting efficiency and therapeutic outcomes, representing a significant step forward in OA management. Further research is needed to optimize these modifications for clinical applications and achieve a balance between therapeutic efficacy and safety [67].

The literature shows massive development of nanomaterials and biodegradable delivery systems for improving the efficacy of OA. Currently, developed material systems encompass carbon-based materials, polymeric materials, protein nanoparticles, organic liposomes, micelles, and dendrimers [76,77,78]. Chondrocytes, the sole cellular inhabitants of cartilage, are responsible for the maintenance and synthesis of the cartilage ECM and represent key targets for drug delivery to improve treatment efficacy. A variety of drugs targeting chondrocytes have been developed, including nucleic acids, phages, growth factors, chondroprotectors, and matrix metalloproteinase inhibitors [79,80]. The uptake of nanomedicines by chondrocytes can be enhanced using cell-penetrating peptide surface modifications. The binding of cell-penetrating peptide-modified nanocarriers to chondrocytes is significantly higher than that of unmodified nanocarriers. Hyaluronic acid (HA) modification can enhance the affinity of nanocarriers for chondrocytes. Cell uptake experiments have demonstrated that this increased affinity primarily results from the binding of HA to the CD44 receptors on the chondrocyte surface, which facilitates the internalization of nanoparticles (NPs). Active targeting of the synovial membrane is mainly directed at macrophages [81,82].

Theranostic nanoparticles, such as those designed for responsive drug release upon reactive oxygen species (ROS) exposure, offer diagnostic and therapeutic capabilities. For example, TKCP@DEX nanoparticles have been shown to effectively release dexamethasone and mitigate cartilage damage in OA [83]. Nanotechnology also facilitates the localized delivery of bioactive agents, minimizing systemic side effects and enhancing cartilage regeneration [84]. Emerging nanotechnology-based gene therapies utilizing nanoparticles, liposomes, or viral vectors are proving effective for localized delivery of therapeutic genes and non-coding RNAs, further improving OA treatment [85,86].

Nanotechnology-enhanced biomaterials stimulate tissue regeneration by supporting cellular responses and enhancing cartilage growth. Green nanomaterials derived from renewable natural sources are being explored for their biocompatibility and environmental benefits, representing a promising direction for cell-targeted OA therapies [87,88,89].

#### 2.4.3. Influence of Nanoparticle Properties on Cartilage Penetration

The size and shape of nanomaterials are critical factors in determining their ability to penetrate biological tissues, including cartilage. Studies have shown that nanoparticles with diameters in the tens of nanometers range (typically 10–100 nm) can more easily infiltrate the dense cartilage matrix due to their ability to pass through nanoscale pores and interact with the ECM [90]. However, particle size is not the only factor affecting cartilage penetration; surface properties, such as charge and hydrophobicity, also play a significant role in the efficiency of nanoparticle diffusion [91].

Surface charge and hydrophobicity significantly influence nanoparticle penetration and distribution in cartilage tissue [92]. Positively charged nanoparticles exhibit stronger binding to the negatively charged cartilage ECM due to electrostatic attraction [93]. However, this strong affinity may lead to increased cytotoxicity or hinder diffusion into deeper cartilage layers [91]. Conversely, modifying nanoparticles with hydrophilic polymers like polyethylene glycol (PEG) can improve penetration by reducing aggregation and enhancing distribution [94].

#### 2.4.4. Strategies to Enhance Nanoparticle Retention and Efficacy

The balance between hydrophilicity and hydrophobicity also plays a critical role [95]. Hydrophilic coatings improve biocompatibility and systemic circulation time, while hydrophobic domains facilitate cell membrane penetration, optimizing cartilage targeting. For instance, amphiphilic nanoparticles combining hydrophilic and hydrophobic properties have demonstrated improved penetration and retention within cartilage, as observed in in vitro studies using simulated joint environments [96,97].

Experimental data further reveal that retention time in synovial fluid varies with inflammatory states [91]. For example, nanoparticles coated with PEG retained longer in inflamed synovial fluid due to reduced clearance rates compared to uncoated nanoparticles, highlighting the potential for tailored surface modifications to enhance therapeutic outcomes.

In certain strategies, proteases such as hyaluronidase are used in conjunction with nanoparticles to locally degrade matrix components, thus enhancing nanoparticle infiltration. While this approach can improve penetration, it requires careful optimization to avoid excessive degradation of the cartilage structure, which could lead to tissue damage [65].

Despite the promising potential of cartilage-targeted nanovectors, research in this area is still in its early stages. The lack of large animal model studies or clinical trials means that the efficacy of these vectors remains poorly defined. Further studies are required to optimize vector design, explore the long-term effects, and evaluate the safety and efficacy of these strategies in clinical settings [98].

#### 2.4.5. Advances in Targeted Delivery and Synergistic Therapies

Targeting Inflammatory Factors with Nanotechnology: Nanoparticles have been developed for the targeted delivery of anti-inflammatory agents, particularly those aimed at modulating key inflammatory mediators such as IL-1β and TNF-α. For instance, polymeric nanoparticles functionalized with specific antibodies or ligands enable precise delivery to inflamed tissues. Studies have demonstrated that nanoparticles loaded with IL-1 receptor antagonist (IL-1Ra) significantly reduce synovitis and slow cartilage degeneration [63]. Additionally, gold nanoparticles (AuNPs) have been shown to inhibit IL-1β-mediated chondrocyte apoptosis by modulating the NF-κB signaling pathway, thus improving cartilage health [99].

Delivery of Growth Factors via Nanotechnology: Bone morphogenetic protein-7 (BMP-7), which is a key factor in promoting cartilage regeneration, has seen significant advancements in delivery systems using nanoparticles. Liposomal and hydrogel-based nanoparticle carriers for BMP-7 demonstrate enhanced cartilage repair through sustained release mechanisms. For example, chitosan and hyaluronic acid-modified nanoparticles have been employed to deliver BMP-7 effectively, thereby improving its bioavailability while minimizing side effects [100]. These systems facilitate localized, controlled delivery, thereby enhancing therapeutic outcomes for OA.

Synergistic Effects of Nanoparticles with MSC-Derived Exosomes: The combination of nanoparticles with MSC-derived exosomes presents a novel approach to OA therapy. MSC exosomes contain abundant anti-inflammatory and regenerative molecules, such as miR-140-3p and TGF-β, which are stabilized and potentiated by nanoparticle-based delivery systems. For instance, magnetic nanoparticles loaded with MSC-derived exosomes have demonstrated targeted delivery to cartilage lesions, leading to enhanced therapeutic efficacy [101]. Compared to traditional MSC therapy, this combined approach reduces the risk of immune rejection and provides a multifaceted regulation of OA pathology.

### 2.5. Clinical Applications and Future Directions in Biotherapies and Nanotechnology for OA

The clinical application of regenerative medicine and biotherapy for OA has rapidly progressed over recent years. Emerging therapies aim to repair or regenerate damaged joint tissues, ultimately improving patient symptoms and quality of life.

#### 2.5.1. Current Status of Clinical Trials in Biotherapy and Regenerative Medicine

Recent advances in regenerative medicine and biotherapy have led to many clinical trials focused on treating OA by repairing or regenerating damaged joint tissues. A review of data from ClinicalTrials.gov identified 83 stem cell-based therapy trials for OA, with 29 of these trials completed. These studies examined various MSC sources, including bone marrow, adipose tissue, and umbilical cord-derived MSCs, while investigating factors such as autologous versus allogeneic cell types and varying dosages. Among these, doses ranged from 1 × 10^6^ to 2 × 10^7^ cells per injection, which were administered through intra-articular injections. Control groups commonly included saline or hyaluronic acid to ensure robust comparisons.

Despite the promise of MSC therapy, only 2 of the 29 completed studies integrated scaffolds or matrix materials to support cartilage regeneration, indicating a gap in utilizing biomaterials in clinical applications [102]. Most trials have relied on intra-articular injections to assess efficacy and safety. Current research emphasizes the dual role of regenerative approaches in controlling inflammation and promoting cartilage repair, which is critical to addressing OA pathophysiology. Table 1 summarizes recent clinical trials conducted from January 2021 to August 2024 on stem cell therapy for OA. Numerous clinical trials have established the long-term safety of MSCs; however, assessing their regenerative capacity remains challenging due to several factors. Key contributors to these uncertainties include variability in patient sample sizes, differences in the severity of injury, inconsistency in follow-up criteria, diversity in cell delivery types, adjunctive therapies, and the selection of control groups. Preclinical studies indicate that combining MSCs with biomaterials can enhance regenerative potential and improve cartilage repair outcomes compared to MSCs alone. Nevertheless, despite these advancements, the combined use of stem cells and biomaterials has not yet successfully emulated the properties of native cartilage. This limitation is reflected in the fact that there are currently only two clinical studies investigating cartilage regeneration supported by stem cell-seeded scaffolds or matrix materials, signifying a critical need for further research and innovation in this domain.

#### 2.5.2. Future Perspectives on Nanotechnology in OA Treatment

Nanotechnology has emerged as a promising avenue for developing innovative OA therapies by leveraging the unique properties of nanoparticles to improve drug delivery, enhance therapeutic efficacy, and target specific cellular processes. Here, we synthesize key insights from recent research on the applications of nanotechnology in OA therapy.

**Nanoparticle-Based Drug Delivery Systems:** Various nanoparticle systems have been developed, such as polysaccharide-based nanosystems, which exhibit multifunctional properties to combat OA symptoms by countering inflammation and oxidative stress while enhancing cartilage protection [103]. For instance, garlic-derived exosomes (GDEs) have been shown to alleviate OA progression by modulating the MAPK signaling pathway, demonstrating the potential of nanoparticles in targeting molecular disease mechanisms [104].

**MicroRNAs and EVs:** MicroRNAs (miRNAs) play crucial roles in OA. For example, miRNA-140 is critical for cartilage homeostasis, and EVs derived from MSCs have shown differences in miRNA expression when cultured in two-dimensional versus three-dimensional environments, impacting their therapeutic potential for restoring joint health [105].

**Advanced Biomaterials:** Innovations in biomaterials, such as magnesium oxide nanoparticles and injectable silk fibroin micro/nanospheres, have demonstrated enhanced capabilities as drug carriers, facilitating cartilage repair and reducing oxidative stress [106,107]. These biomaterials support sustained drug release and exhibit properties that improve their stability and therapeutic application in OA.

**Stem Cell Engineering:** Advances in engineering MSCs with nanoparticles to enhance their capacity for cartilage repair have been promising. For example, combining nanoparticles with peptide constructs has improved MSC differentiation and homing in damaged cartilage, providing a significant advantage in cartilage regeneration for OA patients [108]. This dual-targeting strategy shows great promise for improving outcomes in regenerative therapies.

**Targeted Regulatory Mechanisms:** Enhanced strategies have been developed to promote stem cell differentiation and cartilage repair using biocompatible nanoparticle carriers that target specific signaling pathways, such as the PI3K/Akt pathway, which is involved in chondrogenesis [109]. Moreover, combining a stem cell-homing peptide with plasmid DNA has shown efficacy in alleviating OA progression by reducing chondrocyte senescence [110].

Nanotechnology is paving the way for the development of advanced OA therapies that address conventional treatments’ limitations. Emerging research underscores the multifaceted applications of nanoparticles, including enhanced drug delivery systems, targeted stem cell therapies, and advanced biomaterials aimed at cartilage repair. By effectively targeting the underlying molecular mechanisms of OA and optimizing therapeutic delivery methods, these innovations hold immense potential for improving outcomes in OA management. Continued research and clinical translation of these advances in nanotechnology promise to revolutionize OA treatment, ultimately enhancing the quality of life for patients suffering from this chronic condition.

To overcome the limitations of preclinical studies that primarily rely on small animal models, the integration of large animal models, such as canine and equine systems, provides critical translational insights due to their anatomical and biomechanical similarities to human joints. Canine OA models have demonstrated the efficacy of nanoparticle-based therapies in mitigating cartilage degradation and modulating inflammatory responses [72]. Similarly, equine models offer valuable data on nanoparticle distribution and clearance dynamics under conditions of high joint mobility, which closely mimic human activity [111,112]. These findings underscore the importance of optimizing nanoparticle surface modifications to enhance tissue penetration and retention while minimizing potential adverse effects [98]. Furthermore, preliminary clinical trials have shown promising outcomes with PEG-coated nanoparticles and RGD-functionalized delivery systems, which improved joint function and reduced inflammatory markers [63,113]. However, these results require further validation in larger patient cohorts to establish their clinical utility.

Moving forward, expanding the use of large animal models, developing standardized clinical trial protocols, and bridging the gaps between preclinical and clinical research will be essential steps toward the successful clinical translation of nanotechnology-based OA therapies. Understanding the molecular mechanisms underlying nanoparticle-cartilage interactions and integrating multifunctional therapeutic strategies combining nanotechnology and MSCs hold the potential to revolutionize OA treatment. These innovations offer hope for enhanced outcomes and improved quality of life for patients suffering from this chronic condition.

## 3. Conclusions

In conclusion, stem cell and gene therapies represent significant advancements in biotherapy for cartilage repair in OA. When combined with nanomaterials, these therapies can further enhance treatment targeting and efficiency, providing new hope for improved outcomes. However, it is important to acknowledge that monotherapy research is still primarily at the animal and molecular research stages, lacking robust clinical validation and mechanistic confirmation. Future research should focus on understanding these therapies’ long-term effects and potential risks, particularly in the context of clinical applications.

Clinically, it is essential to prioritize the safety and efficacy of these therapies through well-designed human trials to address the current gaps in clinical data. This approach will enable the further development of biotherapies and regenerative medicine, ultimately advancing the treatment options for OA and improving patient quality of life.

## Figures and Tables

**Figure 1 ijms-25-13384-f001:**
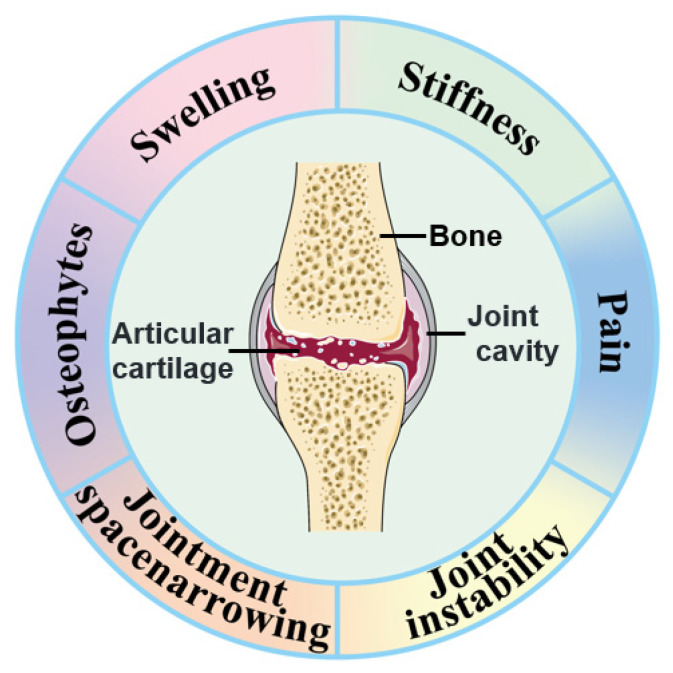
Clinical manifestations of OA.

**Figure 2 ijms-25-13384-f002:**
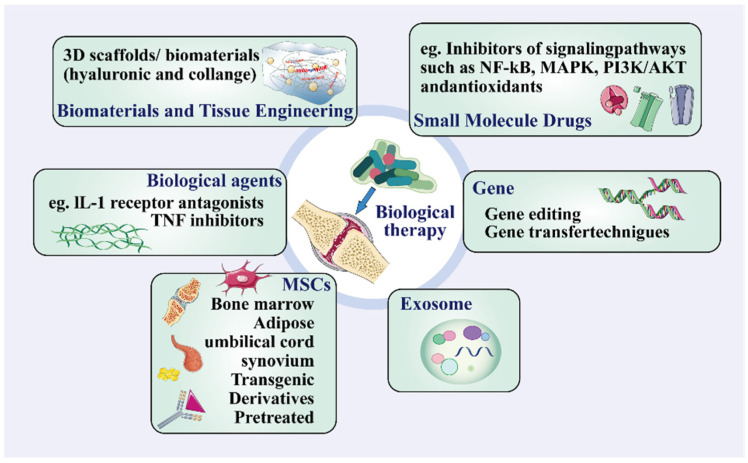
Biological treatment of OA. Biological therapy for OA refers to the use of biologics or bioengineering methods to treat OA. These therapies can include biological products such as biological agents, growth factors, stem cell therapy, gene therapy, and others.

**Figure 3 ijms-25-13384-f003:**
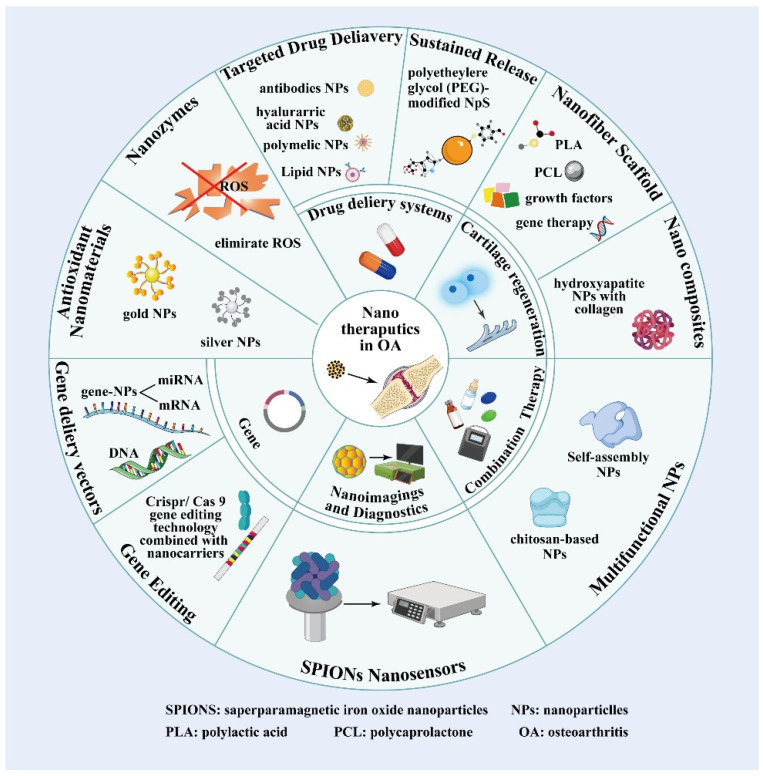
Application of nanotechnology in OA.

**Table 1 ijms-25-13384-t001:** Twenty-nine studies on the use of stem cells in the treatment of OA from January 2021 to August 2024.

Trial Name/ID	Study Population	Administration Route	Stem Cell Source or Derived	Study Status	Official Title
Trial 1	Knee OA patients	Intra-articular Injection	UCB-MNCs	Completed	Clinical Study of Cord Blood Mononuclear Cells (UCB-MNCs) in the Treatment of Knee OA
Trial 2	Knee OA patients	Intra-articular Injection	Unknown	Unknown	Cell Therapy for Patients With Symptomatic Knee OA: Phase I/II, Controlled, Randomized and Double-blind Clinical Trial
Trial 3	OA patients	Intra-articular Injection	BM-MSCs and Lip-MSCs	recruiting	A Phase 2, Randomized Study to Compare Bone Marrow Aspirate Versus Lipoaspirate Concentrate Autologous Cell Therapy for the Treatment of the Knee and Hip OA in Adults
Trial 4	OA patients	Intra-articular Injection	AT-MSCs	Completed	Clinical Trial to Evaluate Safety and Efficacy of MesoCellA-Ortho Tissue-Engineered Advanced Therapy Product in Patients With Osteoarthrosis and Civilisation Diseases
Trial 5	OA patients	Intra-articular Injection	MSC-derived exosomes	recruiting	The Efficacy of Allogenic MSCs Derived Exosomes in OA Patients
Trial 6	OA patients	Intra-articular Injection	UC-MSCs	recruiting	Mechanisms of Treatment Effects Using Cultured, Allogeneic Mesenchymal Stromal Stem Cells (MSCs) in Knee OA
Trial 7	OA patients	Intra-articular Injection and intravenous infusion	UC-MSCs	recruiting	Safety of Cultured Allogeneic Adult Umbilical Cord Derived MSCs for the Treatment of OA
Trial 8	OA patients	Intra-articular Injection	UC-MSCs	recruiting	A Phase 1, Open Label, Dose Escalation Study to Evaluate the Safety and Tolerability After Intra Articular (IA) Injection of UMC119-06-05 in Adult Subjects With Mild to Moderate Knee OA
Trial 9	ex vivo OA model	culture medium	ADSCs	recruiting	Secretome From Mesenchymal Stem/Stromal Cells on Human Osteochondral Explants: Cocktail of Factors Secreted by Adipose-derived Stromal Cells (ASC) for the Treatment of OA and/or for Articular Regeneration
Trial 10	OA patients	Intra-articular Injection	Unknown	Completed	Effects of Muscle Energy Technique Along Conventional Physical Therapy After MSC Transplantation in Knee OA Patients
Trial 11	OA patients	Intra-articular Injection	ADSCs	Not yet recruiting	A Multicenter, Randomized, Double-blind, Controlled Phase III Trial of Allogenic Adipose Tissue-Derived Mesenchymal Progenitor Cells (AlloJoin^®^) Therapy for Knee OA
Trial 12	OA patients	Intra-articular Injection	Amniotic Fluid MSCs (AFCC)	Enrolling by invitation	A Pilot Study: Safety and Efficacy of Allogenic MSC Type AFCC for Treating in Elderly Knee OA Patients
Trial 13	OA patients	Intra-articular Injection	ADSCs	Not yet recruiting	A Multicenter, Randomized, Double-blind, Controlled Phase III Trial of Allogenic Adipose Tissue-Derived Mesenchymal Progenitor Cells (AlloJoin^®^) Therapy for Knee OA
Trial 14	OA patients	Intra-articular Injection	ADSCs	Not yet recruiting	Long-Term Extension Study of Investigator Initiated Trial to Evaluate Cartilage Regeneration by Arthroscopy in Patients With K-L Grade III Knee OA After [JOINTSTEM] Administration
Trial 15	OA patients	Intra-articular Injection	AT-MSCs	Recruiting	Effect of Autologous Adipose Tissue-derived MSCs Therapy in Cartilage Regeneration Among Individuals With Primary Knee
Trial 16	OA patients	Intra-articular Injection	UC-MSCs	Unknown	Human Umbilical Cord MSCs in the Treatment of Knee OA
Trial 17	OA patients	Intra-articular Injection	AD-MSCs	Recruiting	Allogenic MSC Intraarticular Injection for Knee OA Therapy, an RCT Explorative Mode-of-action Study
Trial 18	OA patients	Intra-articular Injection	AD-MSCs	Not yet Recruiting	A Disease-based Treatment Study for Diagnosed OA Utilizing Adipose-derived Regenerative Cells
Trial 19	OA patients	Intra-articular Injection	UC-MSCs	Recruiting	Study on the Safety and Tolerance of MSCs Mediated by Arthroscopy in Patients With OA
Trial 20	OA patients	Intra-articular Injection	Autologous MSCs and Allogenic BM-MSCs	Not yet recruiting	Phase III, Multicenter, Randomized, Open-label Clinical Trial Comparing Treatment With Allogeneic Mesenchymal Cells Versus Autologous Mesenchymal Cells and Versus Active Control With Hyaluronic Acid in Patients With Knee OA
Trial 21	OA patients	Intra-articular Injection	Unknown	Recruiting	Effectiveness of Physical Therapy on Stem Cell Transplantation Recipients in Improving Pain, Quadriceps Muscle Strength and Functional Status of Knee OA: A Randomized Controlled Trial
Trial 22	OA patients	Intra-articular Injection	PRP and UC-MSCs	Completed	Effectiveness of PRP, Conditioned Medium UC-MSCs Secretome and Hyaluronic Acid for the Treatment of Knee OA
Trial 23	OA patients	Intra-articular Injection	MSCs derivatives	Recruiting	Development of Biomedical Technology for the Treatment of Ankle Cartilage Using Injectable Biocomposite Hydrogel
Trial 24	OA patients	Intra-articular Injection	NCR	Not yet recruiting	A Trial to Evaluate the Safety, Tolerability, and Efficacy of NCR100 Injection in the Treatment of Subjects With KOA
Trial 25	OA patients	Intra-articular Injection	XSTEM-OA	not recruiting	A First-in-Human Study of XSTEM-OA in Patients With Knee OA
Trial 26	OA patients	Intra-articular Injection	UC-MSCs	Not yet recruiting	Enabling Advanced Medical Therapy in the Americas: A Pilot Study to Validate a “Ready-to-use” Intra-articular Formulation of Mesenchymal Stromal Cells, Aiming Toward an Effective and Scalable Treatment for Symptomatic Knee OA
Trial 27	OA patients	Intra-articular Injection	FURESTEM-OA Kit Inj	Not yet recruiting	A Single, Dose Escalation, Optimal Dose Finding Phase I/IIa Clinical Trial to Evaluate Safety and Explore Efficacy of the Single Treatment of FURESTEM-OA Kit Inj. in Patients With Knee OA
Trial 28	OA patients	Intra-articular Injection	UC-MSCs	Recruiting	Single Center, Open-label, Dose-increasing Phase I Clinical Trial of UC-MSCs for the Treatment of Knee OA
Trial 29	OA patients	Intra-articular Injection	BMAC	Completed	Autologous Bone Marrow Aspirate Concentrate Injection for the Treatment of Early OA: a Randomized Controlled Trial, Ibn-Sina Hospital, Baghdad 2022

Note: The abovementioned data are all sourced from ClinicalTrials.gov; OA: osteoarthritis; MSC: mesenchymal stem cells; XSTEM-OA: allogeneic adipose tissue-derived integrin α10β1-selected and expanded mesenchymal stem cell (MSC) product; UCB-MNCs: cord blood mononuclear cells; NCR: mesenchymal stromal cells; PRP: platelet-rich plasma; AD-MSCs: allogenic adipose-derived mesenchymal stem cells; BM-MSCs: bone marrow-derived mesenchymal stem cells; BMAC: bone marrow aspirate concentrate; AT-MSCs: autologous adipose tissue-derived mesenchymal stem cells; ADSCs: allogenic adipose tissue-derived mesenchymal progenitor cells.

## Data Availability

Not applicable.

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
