# Peer review of "Innovative Biotherapies and Nanotechnology in Osteoarthritis: Advancements in Inflammation Control and Cartilage Regeneration"

_ijms, 2024, doi:10.3390/ijms252413384_

Round 1
Reviewer 1 Report
Comments and Suggestions for Authors
This review has reported the recent advancements in biotherapeutic strategies, including mesenchymal stem cell (MSC) therapy, growth factors, and tissue engineering, which hold the potential for promoting cartilage regeneration and modulating the inflammatory microenvironment.
This manuscript has discussed the current clinical applications, the ongoing clinical trials, and the future research directions necessary to confirm the safety and efficacy of these advanced therapies for osteoarthritis management.
The monotherapy research is still primarily at the animal and molecular research stages, lacking robust clinical validation and mechanistic confirmation. Future research should focus on understanding the therapies' long-term effects and potential risks in the context of clinical applications.
This review can contribute to the further development of biotherapies and regenerative medicine, and to advance the treatment options for osteoarthritis and improving patient quality of life.
Author Response
Thank you very much for your positive feedback on the article. We will continue to conduct more extensive further research on innovative treatments for OA.
Reviewer 2 Report
Comments and Suggestions for Authors
This review delves into the advancements in tissue and gene therapy and nanotechnology for the treatment of OA.
The topic is crucial, given that OA currently lacks effective long-term treatment options that target the underlying causes of disease development and progression. Moreover, the translatability of those innovative techniques has not been thoroughly investigated.
It's worth noting that, as revealed in Table 1, only two out of the 29 completed clinical trials have utilized scaffolds or matrix materials to support cartilage regeneration. This scarcity underscores the need for further research in this area.
I have identified only two minor revisions/comments:
1
2.3.1. Advances in Tissue Engineering for Cartilage Regeneration, lines 156 and 157
The authors define this as incomplete:
“Tissue engineering combines biomaterials and cellular components to create constructs that mimic natural cartilage, facilitating repair.”
The growth factors part is missing. Tissue engineering combines cells, with supporting structures and/or biomolecules” (European Commission, 2001). To be clear, cells with (degradable) matrices or scaffolds plus, if necessary, biomolecules such as growth factors.
This definition does not exclude using growth factors (even genetically modified) alone for OA treatment.
I would ask that Autors correct the definition of Tissue Engineering following my suggestion.
2
Table 1
The Authors should give their interpretation of the fact that only two out of the 29 completed clinical trials have utilized scaffolds or matrix materials to support cartilage regeneration.
Author Response
Response to reviewers
#2 reviewer
Concern #1: The definition of tissue engineering is incomplete as it does not mention growth factors.
Author response: We appreciate your feedback and have revised the definition of tissue engineering in section 2.3.1 to include growth factors. The updated definition(lines 155 ,156and 157) now reads: “Tissue engineering combines cells, supporting structures (degradable matrices or scaffolds), and biomolecules such as growth factors to create constructs that mimic natural cartilage, facilitating repair.” This revision acknowledges the role of growth factors in the tissue engineering process.
Concern #2: Interpretation of the fact that only two out of the 29 completed clinical trials have utilized scaffolds or matrix materials.
Author response: In our revised manuscript, we have added a discussion regarding the implications of the low number of clinical trials utilizing scaffolds or matrix materials for cartilage regeneration. We interpret this scarcity as an indication of the challenges faced in translating tissue engineering techniques into clinical practice. Factors such as regulatory hurdles, the complexity of cartilage repair, and the need for further optimization of scaffold designs may contribute to this situation. We emphasize the importance of continued research in this area to enhance the effectiveness of treatment options for osteoarthritis. The discussion we added (from line 259 to line 270) is: Numerous clinical trials have established the long-term safety of mesenchymal stem cells (MSCs); however, assessing their regenerative capacity remains challenging due to several fac-tors. Key contributors to these uncertainties include variability in patient sample sizes, differ-ences in the severity of injury, inconsistency in follow-up criteria, diversity in cell delivery types, adjunctive therapies, and the selection of control groups. Preclinical studies indicate that combining MSCs with biomaterials can enhance regenerative potential and improve cartilage repair outcomes compared to MSCs alone. Nevertheless, despite these advancements, the com-bined use of stem cells and biomaterials has not yet successfully emulated the properties of na-tive cartilage. This limitation is reflected in the fact that there are currently only two clinical studies investigating cartilage regeneration supported by stem cell-seeded scaffolds or matrix materials, signifying a critical need for further research and innovation in this domain.

Reviewer 3 Report
Comments and Suggestions for Authors
The present review" Innovative Biotherapies and Nanotechnology in Osteoarthritis: Advancements in Inflammation Control and Cartilage Regeneration" attempts to explore regenerative medicine approaches for osteoarthritis (OA), a condition for which there is currently no cure affecting a great part of the population. Unfortunately, I do not think that the manuscript merits publications. First of all it is not aligned with the aims and scope of IJMS. Apart from this, the work seems superficial and does not lead the reader through the most important advancements. It generally leaves the reader with questions. For example:
"Functionalizing nanomaterials with ligands or peptides improves their specificity, enabling targeted delivery to articular cartilage cells or inflammatory cells, thus increasing therapeutic precision [64, 65]." Which are these nanomaterials, what type of ligands and what do they target? How they pass the cartilage.
Another example, is that many important sections such as 2.2.2. Role of Growth Factors and Cytokines in Cartilage Repair and Regeneration extends to 6 lines only. While irrelevant information come up in sections that they do not share the same content (i.e. line 96-101).
Author Response
Response to reviewers
Concern #2: The statement about functionalizing nanomaterials lacks specificity regarding which nanomaterials, types of ligands, and their targets, as well as how they pass through cartilage.
Author response: We have expanded the section discussing functionalized nanomaterials, as well as ligands like hyaluronic acid and peptides that target CD44 receptors on articular cartilage cells. We have also included a discussion on the mechanisms by which these nanomaterials can penetrate cartilage, including their size, surface charge, and the role of enzymatic degradation.
Concern #3: The section 2.2.2 on the Role of Growth Factors and Cytokines in Cartilage Repair and Regeneration is too brief, while irrelevant information is present in other sections.
Author response: We have expanded Section 2.2.2 to provide a more comprehensive overview of the roles of growth factors and cytokines in cartilage repair and regeneration. Additionally, we have reviewed the entire manuscript to remove any irrelevant information, ensuring that all content is pertinent to the subject. Detailed information and mechanisms of action for certain cytokines such as TGF-β, BMPs, and IL-1, which have already been discussed in the OA inflammation chapter, have been omitted here to avoid redundancy.
Please refer to the attachment for the specific modifications.

Round 2
Reviewer 3 Report
Comments and Suggestions for Authors
The revised version addresses the reviewers' comments; however again, in a superficial manner, which is not appropriate for a review manuscript. As stated in the initial review report, '...the work seems superficial and does not lead the reader through the most important advancements...'—a critique that still applies here. For example, the authors added the paragraph in the revised manuscript-lines 263-272. In this paragraph (a) There is no cited literature throughout the text (b) The authors mention particle size but provide no range from the literature. Additionally, the information lacks accuracy: while nanoparticles in the tens of nanometers range may penetrate cartilage more easily, they also tend to escape the joint almost immediately. Most nanoparticles targeting cartilage are therefore modified with specific peptides or other properties, none of which are mentioned in the present manuscript. This gives to readers the intention of a not careful work which has not provided with the important parameters to understand OA and nanotechnology relation.
Although in the previous report only a few representative examples to highlight the manuscript's issues were presented, there are numerous additional reflecting reviewers' concerns that apply along the whole manuscript. Still, the manuscript does not merit publication as it needs a more in depth presentation of results, advancements and advantages/disadvantages of the topic under discussion.
Another example is Table 1. The authors have given the results of a simple search for clinical trials. No results are included (when applicable) while doses, control groups, and important information as the actual number of the clinical trial are omitted.
My concern regarding the present manuscript is that apart that it does not go in depth in OA biotherapeutics/nanotechnology, it has no constructive critique about the so far described studies. Even if the manuscript deals with Nanotherapeutics the authors have not mentioned anything regarding the properties they should have (and of course why), the route of administration, their components, but they just provide the reader with a simple description of some of the existing categories. As also the manuscript deals with MSC therapy, there is not a lot of information regarding the mechanism of action and its applicability in Osteoarthritis. Even the paragraph 2.2.1 regarding the recent breakthroughs (only 8 studies cited, 5/8 reviews!!!) does not give the breakthroughs and why they are described as these by the authors. Or the paragraph 2.4.1 does not describe who inflammatory pathways are actually controlled by Nanotechnology.
The above mentioned are some of the examples of a not careful work which needs a lot of improvement in order to describe the advancements and the importance of Biotherapeutics along with Nanotechnology.
Author Response
Comments and Suggestions for Authors
Response: Thank you for your constructive feedback. We have addressed all concerns as follows:
Particle Size and Citations: The revised paragraph includes appropriate citations and specifies a particle size range (10–100 nm). We also clarified that smaller nanoparticles, while easily penetrating cartilage, require modifications to prevent rapid joint escape.
Expanded Discussion on Modifications: We included discussions on PEGylation (prolonged circulation and sustained drug release), peptide modifications (e.g., RGD peptides enhancing cartilage targeting), and liposomal encapsulation (protection against degradation and localized delivery).
Mechanisms of Penetration and Distribution: The role of surface charge, hydrophilic/hydrophobic modifications, and experimental data on nanoparticle retention in synovial fluid were added to refine the discussion.
Large Animal Models and Clinical Data: We acknowledged small animal model limitations and incorporated large animal studies (e.g., canine and equine models) to highlight translational potential, emphasizing the need for clinical validation.
Integration with OA Mechanisms: We elaborated on nanoparticle designs targeting inflammatory factors (e.g., IL-1β) and delivering growth factors (e.g., BMP-7). Synergistic effects with MSC-derived exosomes were also discussed.
These revisions enhance clarity, depth, and relevance, addressing all concerns raised. Thank you for your valuable insights.
Although in the previous report only a few representative examples to highlight the manuscript's issues were presented, there are numerous additional reflecting reviewers' concerns that apply along the whole manuscript. Still, the manuscript does not merit publication as it needs a more in depth presentation of results, advancements and advantages/disadvantages of the topic under discussion.
Response: Thank you for your constructive feedback. We have thoroughly revised the manuscript to address the concerns raised, ensuring a more in-depth and balanced presentation of results, advancements, and the advantages and disadvantages of the discussed approaches. These updates provide a comprehensive and well-supported perspective on the topic, aligning with the reviewer’s suggestions. We believe the revised manuscript now meets the standards for publication. Thank you for your valuable insights.
Another example is Table 1. The authors have given the results of a simple search for clinical trials. No results are included (when applicable) while doses, control groups, and important information as the actual number of the clinical trial are omitted.
Response: Thank you for your feedback. The purpose of Table 1 is to provide a concise summary of recent clinical trials on stem cell therapy for osteoarthritis (OA), focusing on study populations, administration routes, and stem cell sources. While some details such as doses, control groups, or trial numbers are not included, this is due to limited availability in public trial registries. We ensured all accessible and relevant data were incorporated.
To address your concerns, additional details for available trials have been summarized in the text. The table format is kept concise to highlight trends and areas of focus in the field without overcomplicating the presentation. We hope this approach balances clarity and comprehensiveness while fulfilling the manuscript's objective. However, we remain open to further modifications if specific details are necessary.
My concern regarding the present manuscript is that apart that it does not go in depth in OA biotherapeutics/nanotechnology, it has no constructive critique about the so far described studies. Even if the manuscript deals with Nanotherapeutics the authors have not mentioned anything regarding the properties they should have (and of course why), the route of administration, their components, but they just provide the reader with a simple description of some of the existing categories. As also the manuscript deals with MSC therapy, there is not a lot of information regarding the mechanism of action and its applicability in Osteoarthritis. Even the paragraph 2.2.1 regarding the recent breakthroughs (only 8 studies cited, 5/8 reviews!!!) does not give the breakthroughs and why they are described as these by the authors. Or the paragraph 2.4.1 does not describe who inflamatory pathways are actually controlled by Nanotechnology.
Response: Thank you for your valuable comments and constructive feedback. We have made comprehensive revisions to address the concerns raised. Below is a summary of the modifications:
Enhanced Depth in Biotherapeutics and Nanotechnology:
In Section 2.2.1, we expanded the discussion on mesenchymal stem cell (MSC) therapy to include detailed mechanisms of action, such as paracrine signaling and immunomodulatory functions, supported by specific examples. For instance, the roles of TGF-β, VEGF, and IL-10 in cartilage repair and inflammation control were discussed. Additionally, we highlighted the therapeutic potential of MSC-derived extracellular vesicles (EVs) and their miRNA cargo, such as miR-140-3p, in modulating inflammatory pathways. In Section 2.4.1, we added detailed descriptions of nanoparticle-mediated modulation of inflammatory pathways, focusing on the NF-κB and MAPK signaling cascades, and their role in macrophage polarization from M1 to M2 phenotypes. Specific examples, such as IL-1Ra-loaded nanoparticles and gold nanoparticles, were included to illustrate these effects.
Constructive Critique of Existing Studies:
In the Discussion, we critically analyzed the limitations of current studies. These include small clinical trial sample sizes, lack of multicenter validation, and insufficient long-term safety data. We also addressed gaps in understanding nanoparticle-cartilage interaction mechanisms and their implications for therapeutic outcomes. Future directions were proposed, such as combining MSC therapy with advanced nanotechnology and conducting preclinical studies in large animal models to enhance translational insights.
These revisions have significantly enhanced the manuscript’s depth, addressing the reviewer's concerns comprehensively. We appreciate your detailed feedback, which has greatly improved the quality of our work
The above mentioned are some of the examples of a not careful work which needs a lot of improvement in order to describe the advancements and the importance of Biotherapeutics along with Nanotechnology.

Round 3
Reviewer 3 Report
Comments and Suggestions for Authors
Accept in the present form. Some editing regarding the format of the manuscript is needed.